# Individualized Bleeding Risk Assessment through Thromboelastography: A Case Report of May–Hegglin Anomaly in Preterm Twin Neonates

**DOI:** 10.3390/children8100878

**Published:** 2021-10-01

**Authors:** Ilaria Amodeo, Genny Raffaeli, Federica Vianello, Giacomo Cavallaro, Valeria Cortesi, Francesca Manzoni, Giacomo S. Amelio, Silvia Gulden, Fabio Mosca, Stefano Ghirardello

**Affiliations:** 1Neonatal Intensive Care Unit, Fondazione IRCCS Ca’ Granda Ospedale Maggiore Policlinico, 20122 Milan, Italy; amodeoilaria@gmail.com (I.A.); genny.raffaeli@unimi.it (G.R.); federica.vianello@policlinico.mi.it (F.V.); valeria.cortesi92@gmail.com (V.C.); francesca.manzoni1@unimi.it (F.M.); giacomo.amelio3@gmail.com (G.S.A.); silvia.gulden@hotmail.it (S.G.); fabio.mosca@unimi.it (F.M.); s.ghirardello@smatteo.pv.it (S.G.); 2Department of Clinical Sciences and Community Health, Università degli Studi di Milano, 20122 Milan, Italy; 3Pediatric Nephrology, Dialysis and Transplant Unit, Fondazione IRCCS Ca’ Granda Ospedale Maggiore Policlinico, 20122 Milan, Italy; 4Neonatal Intensive Care Unit, Fondazione IRCCS Policlinico San Matteo, 27100 Pavia, Italy

**Keywords:** May–Hegglin anomaly, MYH9-related disorders, congenital thrombocytopenia, macrothrombocytopenia, viscoelastic test

## Abstract

May–Hegglin anomaly (MHA) is a rare autosomal dominant disorder in the spectrum of myosin heavy chain-related disorders (MYH9-RD), characterized by congenital macrothrombocytopenia and white blood cell inclusions. MHA carries a potential risk of hemorrhagic complications. Bleeding diathesis is usually mild, but sporadic, life-threatening events have been reported. Data regarding the clinical course and outcomes of neonatal MYH9-RD are limited, and specific guidelines on platelet transfusion in asymptomatic patients are lacking. We present monochorionic twins born preterm at 32 weeks of gestation to an MHA mother; both presented with severe thrombocytopenia at birth. Peripheral blood smear demonstrated the presence of macrothrombocytes, and immunofluorescence confirmed the diagnosis of MHA. Close clinical monitoring excluded bleeding complications, and serial hemostatic assessments through a viscoelastic system demonstrated functionally normal primary hemostasis in both patients. Therefore, prophylactic platelet transfusions were avoided. Whole DNA sequencing confirmed the pathogenetic variant of MHA of maternal origin in both twins. Thromboelastography allowed real-time bedside bleeding risk assessment and supported individualized transfusion management in preterm newborns at risk of hemostatic impairment. This report suggests that dynamic and appropriate clotting monitoring may contribute to the more rational use of platelets’ transfusions while preserving patients with hemorrhagic complications and potential transfusion-related side effects.

## 1. Introduction

May–Hegglin anomaly (MHA) is a rare autosomal dominant disorder resulting from mutations in the MYH9 gene located on chromosome 22q12.3, coding for the nonmuscle myosin heavy chain IIA (NMMHC-IIA) [1]. Different mutations result in allelic variants and distinct phenotypic syndromes referred to as myosin heavy chain-related disorders (MYH9-RD) [2]. MYH9-RD is a spectrum of autosomal dominant disorders that share platelet macrocytosis, thrombocytopenia, and leukocyte inclusion bodies at variable degrees. In addition, MYH9-RD is the most frequently inherited form of thrombocytopenia, but its precise incidence is unknown [3].

A possible combination of thrombocytopenia and altered platelet function in the neonatal period could lead to impaired hemostasis, bleeding, and the need for transfusions, with increased morbidity [4]. However, data from neonatal MYH9-RD are scarce, and no specific guidelines for platelet transfusion in asymptomatic newborns with this genetic condition are available.

We describe the case of two premature twins born to an MHA mother who presented with severe asymptomatic thrombocytopenia at birth. We discuss how clinical monitoring and evaluation of hemostasis through serial viscoelastic tests guided an individualized bleeding risk assessment and transfusion approach by reporting this case.

## 2. Case Presentation

Parents provided written informed consent for the genetic analysis test and the publication of this case report, including images. All information revealing the subject’s identity was avoided, and all information was anonymized. A 32-year-old gravida 1 para 0 woman was affected by MHA secondary to a heterozygous mutation in the MYH9 gene (p.Val34glu). She had spontaneous monochorionic diamniotic twin gestation. The parents were not consanguineous. Pregnancy was characterized by intrauterine growth restriction of twin 1 and cerebral anomalies of both twins on obstetric ultrasound. Fetal magnetic resonance imaging (MRI) showed mild ventriculomegaly associated with a cyst in the velum interpositum. Emergency cesarean section was performed at 32^+1^ weeks of gestation due to altered umbilical and medial cerebral artery flowmetry of the second fetus.

At birth, they required routine resuscitation and were then admitted to our neonatal intensive care unit (NICU) because of prematurity and moderate respiratory distress. The APGAR score was 6^1′^/9^5′^ for twin 1 and 5^1′^/7^5′^ for twin 2. Both were male and weighed 1490 g and 2010 g, respectively. Twin 1 required noninvasive respiratory support during the first day of life, while twin 2 required one surfactant administration on day 1 and continued noninvasive respiratory support until day 7. The cardiac assessment was normal. Prophylactic antimicrobial therapy with ampicillin and gentamicin was administered during the first 72 h until blood cultures were confirmed to be negative. In addition, a brief cycle of phototherapy was performed for mild hyperbilirubinemia in both twins.

Complete blood count on the first day of life revealed normal red blood cell indices, normal white blood cell count, and differentially associated severe thrombocytopenia (twin 1: 22 × 10^3^ per mm^3^; twin 2: 24 × 10^3^ per mm^3^) without active bleeding. A peripheral blood smear indicated the presence of macrothrombocytes, compatible with the maternal syndrome. The immunofluorescence test was performed on both twins and the mother, confirming the diagnosis of MHA.

The direct platelet (PLT) count on light microscopy dropped to a minimum value of 12 × 10^3^ per mm^3^ on day 3 for twin 1, while it never dropped below 24 x10^3^ per mm^3^ for twin 2. Coagulation assays showed normal prothrombin time (PT) and fibrinogen levels, with a slight increment of activated partial thromboplastin time (aPTT) (maximum aPTT ratio: 1.63 for twin 1, 2.01 for twin 2). Table 1 provides the temporal sequence of hematologic tests performed during the hospital course.

In our NICU, thromboelastography (TEG) is regularly used to assess neonatal hemostasis whenever needed. In the present case, we used a new viscoelastic method (viscoelastic coagulation monitor (VCM^©)^, Entegrion, Durham, NC, USA)). Based on institutional TEG reference ranges, the maximum clot firmness (MCF) values were in the range of age, even at the lowest PLT count (Figure 1).

During the entire hospitalization period, no bleeding complications occurred. As per our unit protocol for premature babies, neonatal cerebral ultrasound was performed daily during the first week of life and then weekly since no hemorrhagic complications arose. Furthermore, based on clinical and thromboelastographic data, PLT transfusions were not required, and the patient was discharged with a stable platelet count (twin 1: 38 × 10^3^ per mm^3^; twin 2: 24 × 10^3^ per mm^3^).

Cerebral MRI confirmed the presence of mild ventriculomegaly and a cyst of the velum interpositum in both twins, without evidence of obstruction of CSF circulation. Therefore, only neurosurgical follow-up is recommended. Eye examination and audiological tests were normal in both twins.

DNA analysis by whole-exome sequencing identified the heterozygous variant p.Val34glu of the MYH9 gene of maternal origin (OMIM *160775). A T-to-A base substitution at position Chr22(GRCh37):g.36745181 (NM_002473.5:c.101T>A) in exon 2 led to the substitution of valine with glutamic acid at position 34 of the protein. It was classified as pathogenetic and associated with the clinical characteristics of the probands. The two newborns were finally included in a multidisciplinary follow-up.

## 3. Discussion

MHA belongs to the MYH9-RD, a spectrum of congenital disorders characterized by a variable degree of macrothrombocytopenia and white blood cell inclusions [3]. A limited spectrum of mutations has been identified so far, and sporadic forms account for approximately 35% of the cases; therefore, the general incidence of this rare condition remains unknown [5].

Nonmuscle class IIA myosins are cytoskeletal contractile proteins found in platelets, T cells, renal cells, and the cochlea. Mutations affecting the head domain (exons 2–19) are associated with severe forms and extra-hematologic involvement, while mutations in the tail domain (exons 21–40) are usually mild, with broad clinical heterogeneity [1,6].

In MHA, impaired cytoskeleton contractility causes abnormal segmentation of megakaryocytes [7]. The distinctive features are macrothrombocytopenia, secondary to ineffective thrombopoiesis, and the presence of Döhle-like blue inclusion bodies in leukocytes resulting from MHC deposition [8]. Platelet numbers vary widely among patients, but they generally remain stable throughout life [1]. Electronic instruments can underestimate PLT counts up to 10-fold lower values; therefore, microscopic counting is essential to assess bleeding risk [1,2]. In addition, an immunofluorescence test for NMMHC-IIA distribution within neutrophils is detrimental [1]. Finally, molecular studies of the MYH9 gene should be performed to confirm the diagnosis and allow a genotype–phenotype correlation, which is essential for prognostic purposes [1,5].

MYH9-RD should be suspected in any case of congenital macrothrombocytopenia. However, de novo mutations are challenging because thrombocytopenia is an incidental finding [1]. Therefore, differential diagnosis is essential to avoid inappropriate and potentially harmful interventions, such as intravenous IgG, corticosteroids, and splenectomy [5].

The main concern in MYH9-RD is related to possible bleeding due to ineffective platelet clot formation. Although it is commonly thought that platelet function is deficient in MHA, aggregation and release reactions tested in vitro, as well as platelet life span, are usually normal or only slightly reduced [9]. A bleeding disorder is usually mild and typically presents in infancy, and its severity remains unchanged over the years. However, some life-threatening events have also been reported [5,8]. Extra-hematologic manifestations usually appear later in life and include bilateral sensorineural hearing loss (60%), proteinuria and progression to renal failure (30%), and presenile cataracts (16%).

Due to its rarity and possible diagnostic delays, data regarding neonatal MYH9-RD are limited; thus, the real risk of bleeding still needs to be determined. Reports of MHA occurring in twins are scarce, and the possible association with cerebral anomalies has not been described thus far [8]. Therefore, although a comparison with previous similar cases is not feasible, multidisciplinary follow-up will identify eventual late-onset complications and contribute to a better knowledge of the disease.

Generally, pregnancy and childbirth are not considered to increase the risk of bleeding in patients with MHA [10]. Pregnant women are not routinely treated to prevent peripartum hemorrhage, and without a significant history of bleeding, vaginal delivery may be managed safely for both mother and child [11]. Although thrombocytopenia may develop in utero, fetuses and newborns do not show increased rates of severe events or intracranial hemorrhages [10,12]. A systematic review of MHA during pregnancy reported 78 live neonates. In total, 34 patients were diagnosed with MHA and thrombocytopenia at birth. Two asymptomatic patients received a prophylactic platelet transfusion soon after birth, and one received blood and platelet transfusions during the third week of life, but no hemorrhagic complications were reported [5].

To date, there is no consensus for prophylaxis or treatment of macrothrombocytopenia or recommended platelet levels to achieve adequate hemostasis in MYH9-RD patients, especially in asymptomatic neonates [13]. Therefore, clinicians usually refer to the general recommendations for neonatal thrombocytopenia [14,15,16,17].

In NICUs, thrombocytopenia affects 18–35% of newborns, approaching 70% of extremely low birth weight infants [18]. A platelet count of <50 × 10^3^ per mm^3^ is most commonly considered a reference transfusion threshold [14,15,16]. Although severity was not shown to correlate with the risk of intraventricular hemorrhage, up to 5–9% of cases receive at least one prophylactic platelet transfusion [16,19,20]. Increasing evidence of adequate primary hemostasis in preterm newborns and the detrimental effects of liberal transfusion are changing clinical practice [21]. Platelets are major effectors of hemostasis, but they are also involved in host immunity, angiogenesis, and inflammation. Major developmental differences exist between neonates, especially in preterm infants and adults [22]. Therefore, the transfusion of adult-derived blood products may disrupt neonatal hemostatic and immune homeostasis due to pro-inflammatory effects [22,23]. A restrictive threshold of 25 × 10^3^ per mm^3^ for platelet transfusion proved to be less harmful than a liberal approach in nonbleeding premature infants [17]. Since MYH9-RD is associated with a mild tendency to bleeding, and PLT transfusions are not without risks, the transfusional approach should be tailored, and global hemostatic assays may support clinical decisions [12].

Indeed, viscoelastic tests, such as thromboelastography (TEG) and thromboelastometry (ROTEM), provide a more accurate picture of the in vivo hemostasis, compared with conventional assays [24]. They are based on the interaction between blood cells and coagulation factors and evaluate the whole-blood clotting process. In short turnaround times, clot kinetics are graphically and numerically represented, from initiation to stabilization and lysis, including the interaction between activated platelets and fibrinogen [25]. In addition, TEG has been shown to reliably assess neonatal hemostasis, and TEG-based reference ranges are available for term and preterm newborns [26,27]. In pediatric cardio surgery, viscoelastic assays proved to optimize blood product use while reducing bleeding [28]. In the neonatal setting, viscoelastic tests have been used to evaluate neonates with bleeding, sepsis, and patent ductus arteriosus under pharmacological treatment [29,30,31].

Within this context of promising clinical applications, a new viscoelastic coagulation monitoring system (VCM^©^, Entegrion, USA) has been recently developed. It is small, portable, user friendly, and can be used at bedside. In addition, it uses limited amounts of native blood (0.3 mL) with rapid initiation of clotting and does not require activating factors, and is particularly suitable for the NICU setting. The VCM^©^ results showed good agreement with the ROTEM NATEM system [32].

In this case, severe thrombocytopenia was detected at birth in both twins. Maternal history facilitated the differential diagnosis. Consequently, a peripheral blood smear and immunofluorescence tests were promptly performed. Close clinical monitoring and serial viscoelastic tests allowed the dynamic assessment of hemostatic competency, despite severe thrombocytopenia and prolonged aPTT.

According to consensus-based international guidelines, both twins should receive platelet transfusion at birth [14,16]. Soon after, the PLT count even dropped under the restrictive threshold for premature newborns, and transfusion would have seemed inevitable. However, VCM^©^ monitoring demonstrated functionally normal hemostasis in both patients. The coagulation cascade was normal, as indicated by the range of clotting time (CT) values. Despite severe thrombocytopenia, the MCF reflects platelet count and function and is layered in the reference range throughout the hospital stay. Functionally competent clotting also depends on fibrinogen levels, but the exact contribution of fibrinogen and platelets cannot be analyzed separately through this test. To address the fibrinogen role, we should have performed the functional fibrinogen TEG, which requires whole blood to add a specific reagent neutralizing platelet function [25]. However, this technical limitation did not affect the clinical management because the interaction between fibrinogen and activated platelets was normal, as indicated by in-range clot formation time (CFT), angle, and amplitude (Figure 1). Thus, prophylactic and unnecessary PLT transfusions were avoided.

Although the identified pathogenic variant carries a high risk of bleeding and extra-hematological involvement, no hemorrhagic diathesis was detected, while renal, audiological, and ophthalmological assessments were normal.

An adequate bleeding risk assessment is critical for therapeutic management and long-term patient outcomes, and this report clearly suggests that viscoelastography holds promise for optimizing platelet administration, even in cases of severe thrombocytopenia. Furthermore, based on the emerging evidence in neonatal transfusion medicine that “less is probably more,” we should implement the use of viscoelastic assays for the hemostatic monitoring of neonates at higher risk of bleeding [17,23,33].

## 4. Conclusions

In conclusion, viscoelastic systems, such as VCM^©^, could guide clinicians in assessing neonatal hemostasis, supporting the individualized transfusional approach. In addition, a more rational and tailored platelet administration will reduce the exposure to potential adverse effects, ultimately contributing to improved neonatal outcomes. This would especially benefit patients at a higher risk of bleeding, such as those affected by MYH9-RD, for which specific guidelines are lacking, particularly in the neonatal period.

## Figures and Tables

**Figure 1 children-08-00878-f001:**
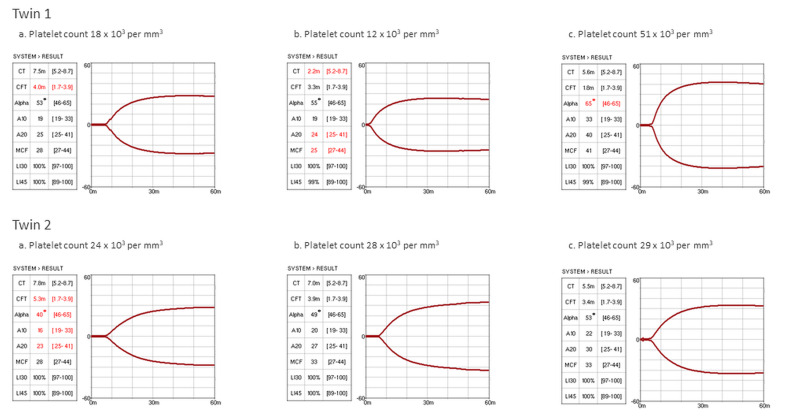
VCM^©^ numerical results and graphic representation of patients’ hemostatic profiles. VCM^©^ analysis showed competent hemostasis despite severe thrombocytopenia in both twins. For each patient, VCM^©^ results and a corresponding platelet count performed at three consecutive times (day 1, day 3, and day 7 for both twins) during hospitalization are shown. CT, clotting time (min); CFT, clot formation time (min); alpha, alpha angle (degrees); A10 and A20, amplitude at 10 and 20 min (VCM units); MCF: maximum clot firmness (VCM units); LI30 and LI45: lysis index at 30 and 45 min (%).

**Table 1 children-08-00878-t001:** Results of complete blood count and differential and conventional hematologic assays at day 1, day 3, and day 7. WBC: white blood cells; RBC: red blood cells; Hb: hemoglobin; Ht: hematocrit; PT: prothrombin time; aPTT: activated partial thromboplastin time.

Laboratory Results	Twin 1			Twin 2		
	Day 1	Day 3	Day 7	Day 1	Day 3	Day 7
WBC (10^9^/L)	8.13	6.58	9	8.6	7.03	11.65
RBC (10^9^/L)	2.7	2.94	2.54	5.37	5.13	5.06
Hb (g/dl)	10.6	11.3	9.1	21.4	20	19.4
Ht (%)	32.4	33.7	27.4	69.9	57.7	55.5
Platelet (10^9^/L)	18	12	51	24	28	29
Fibrinogen (mg/dl)	168	-	270	-	-	154
PT ratio (s)	1.01	-	0.84	-	-	1.17
aPTT ratio (s)	1.63	-	1.2	-	-	2.01

## Data Availability

The data that support the findings of this study are not publicly available because they contain information that could compromise participants’ privacy, but they are available from the corresponding author.

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
