# Peer review of "Individualized Bleeding Risk Assessment through Thromboelastography: A Case Report of May–Hegglin Anomaly in Preterm Twin Neonates"

_children, 2021, doi:10.3390/children8100878_

Round 1

Reviewer 1 Report

The authors reported the clinical course of premature twins suffering from rare MHA disease and the sharing of the experience of thrombelastography to assess the necessity for platelet transfusion in this rare situation. This manuscript describes the treatment process and completes inspection and tracking, providing a good reference for future diagnosis and treatment of related disease cases. 

I have some suggestions for the authors and hope that my comments are constructive to them.

  1. Line 68, please describe the detailed information of flowmetry. From the umbilical vessels or fetal brain vessels?
  2. Line 71. The abbreviation of (B.I.P) and (B.S.). Please clarify it. 
  3. Figure 1. Would you please explain in detail the information in Fig1(a)(b)(c)? Are they the series follow-up data of platelet counts in these twins during hospitalization?
  4. Would you please provide the time point at which you perform neonatal brain sonography to assess intracranial hemorrhage and the values of platelets before discharge during your hospitalization? 

Author Response

Thank you for your suggestions.

  1. Flowmetry of the fetal umbilical artery and medial cerebral artery were evaluated at the prenatal ultrasound, and both were found altered. Therefore, we specified the flowmetry details in the revised manuscript.
  2. I.P. and B.S. were the initials of the baby’s names. However, since we referred to the patients as “twin 1” and “twin 2” throughout the manuscript, we preferred to remove the initials in the revised manuscript.
  3. Figure 1: we specified that figures a,b,c represent consecutive assessments of the hemostatic profile during hospitalization, with the corresponding platelet count.
  4. As requested, we specified the time point for neonatal brain sonography and platelet count before discharge in the case presentation.

Reviewer 2 Report

Thank you for submitting a report on an interesting topic. However, it seems that the following corrections are needed.

Abstract

  1. In the conclusion of the abstract, it would be better to describe the meaning of this report or the facts newly learned through this report rather than the basic (general) story.

Introduction

  1. It seems necessary to further explain Giant Platelet Syndrome and MYH9-RD occurring in newborns.

Case Presentation

  1. It seems necessary to further explain the overall hospital courses of both preterm infants.
  2. It would be nice to add another table or figure in the temporal sequence of the hematologic tests.
  3. Please write down the full name for VCM™ (VISCOELASTIC COAGULATION MONITOR).

Discussion

  1. The main content of the discussion should be the clinical manifestations and managements of May-Hegglin Anomaly in preterm infants or neonates. However, there are not many explanations about this, and only basic (general) contents are described.
  2. The contents are described without a constant flow.
  3. It seems that a little more detailed explanation is needed about the efficacy and the future use of thromboelastography including VCM™ in the NICU.
  4. If there was not enough explanation for Giant Platelet Syndrome and MYH9-RD in the introduction, it seems that additional explanation is needed.

Conclusion

  1. It would be better to describe the meaning of this report or the facts newly learned through this report rather than the basic (general) story.

Thank you.

Author Response

Abstract

  1. Thank you. We modified the abstract as suggested, trying to underline the meaning of this case report.

Introduction

  1. Thank you for your suggestion. We modified the introduction accordingly.

Case Presentation

Thank you.

  1. We further explain the overall hospital courses of the twins, as you can find in the revised manuscript.
  2. We have added Table 1, which reports the temporal sequence of the hematologic tests, as required.
  3. We wrote the full name for VCM, as required

Discussion

Thank you for your suggestions.

1. As we specified in the manuscript, data regarding MHY9-RD and MHA are limited, particularly in the neonatal population (lines 169-171 of the revised manuscript), and this represents the added value of this case report. In the discussion, we first illustrate the general characteristics of MHA, which refer to any period of life, and then we discuss the maternal, fetal, and neonatal manifestations described so far. However, reported data are limited, and the main source of systematic data is represented by the review of Hussein and colleagues, which is cited in the discussion (lines 177-181 of the revised manuscript). Furthermore, we specifically state that “there is no consensus for prophylaxis or treatment of macrothrombocytopenia or recommended platelet levels to achieve adequate hemostasis in MYH9-RD patients, especially in asymptomatic neonates” (lines 182-184 of the revised manuscript). Hence, the importance of reporting this case to underline the need to identify at-risk patients and to individualize the transfusion approach to avoid both bleeding complications and transfusion-related adverse effects (lines 196-199 of the revised manuscript). 

2. We have tried to improve the flow of the discussion ourselves. The external editing service (see certificate) improved the manuscript further for both the English language and the flow. We hope you are happy with this new version. 

3. We have explained the actual clinical application of TEG and VCM and, based on existing increasing knowledge, we have introduced a reflection on the future promising applications of this methodology in transfusion medicine. Finally, we have added references to support these concepts: 22-23; 27-31; 33.

4. Regarding MHY9-RD, we added some more explanation in the introduction, as requested. 

 Conclusion

  1. Thank you for your suggestions. To avoid unnecessary exposure to blood products, we have tried to further underline the promising application of TEG in directing platelet transfusion in a more evidence and rational way. This application is rather new in neonatology, and we think this report may increase clinicians’ awareness in this context. Furthermore, evidence from recent trials (i.e., Planet-2) highly supports the need for neonatologists to improve the transfusion approach.

Thank you.

Round 2

Reviewer 2 Report

Most of the contents pointed out in the first review were well modified and corrected. Authors should check or revise the contents below.

1. Abstract  line 36

platelets -> platelt transfusions ?

2. Case Presentation line 64 & Table 1

The hemoglobin levels of the two monochorionic twins differ by about 10 mg/dL. Did the two monochorionic twins ever be diagnosed with TTTS? Didn’t you conduct a test for complications of TTTS donor and recipient?

Author Response

  1. Abstract  line 36

1. Thank you for your comment. We have modified the text accordingly

  1. Case Presentation line 64 & Table 1

Thank you for your comment. Indeed, there is a discrepancy in Hb values in the two monochorionic twins. However, prenatal ultrasonographic criteria for TTTS were not satisfied, and therefore, the twins have never been diagnosed with TTTS.

In the postnatal period, we monitored the patients and performed multi-organ evaluations. Both twins have remained hemodynamically stable throughout the NICU course. The cardiac evaluation was normal for both of them. The neurological and hematological assessments have been reported in detail in the manuscript.  Although the donor is usually anemic and the recipient is polycythemic, this is not universal. We know that cerebral anomalies can be part of the clinical picture of complicated TTTS pregnancy, but we are not sure that an acute perinatal TTTS may apply to this case. Since TTTS is diagnosed based on ultrasonographic criteria, we have decided and would prefer not to mention it in the report if the reviewer agrees upon it.